# Improving Etched Flatness by Micro Airflow Array Pressurization in ITO Glass Laser Machining

**DOI:** 10.3390/mi14030676

**Published:** 2023-03-19

**Authors:** Rong Chen, Zhaojie Chen, Jin Xie

**Affiliations:** 1College of Mechanical and Electrical Engineering, Guangdong University of Science and Technology, Dongguan 510645, China; rongchen338@163.com; 2School of Mechanical and Automotive Engineering, South China University of Technology, Guangzhou 510640, China

**Keywords:** micro-airflow, air bearing, laser etching, flatness, ITO glass

## Abstract

In laser etching of ITO glass, the warpage due to workpiece positioning causes breakpoint or deformation of micron-scale etching circuits. Based on traditional laser etching, a micro-airflow array pressurization is proposed by using a micro-flow air bearing through airflow positioning. The objective is to achieve high-precision laser etching by pressurized micro-deformation of ITO glass during positioning. First, the micro-air flow and pressurized micro-deformation were modelled in relation to the airflow pressure and etching gap in order to analyze the flatness variation behavior. Then, the surface flatness was investigated in relation to the airflow parameters and relative bearing location. Finally, the critical value of the pressurization parameter were calculated using a data-twin and were applied to industrial ITO glass etching. It is shown that the uniform flow pressure distribution and surface central micro-deformation were formed by positive airflow pressure in the airflow area. The airflow pressure and etching gap could promote surface flatness, while excessive values could result in excessive deformation. Under the micro flow pressure, the initial flatness of the workpiece was able to be compensated within the critical pressurization parameter. By controlling the micro flow stress, the micro-airflow array pressurization could reduce the flatness to 22 μm with stress of 10.7–12.6 Pa. In industrial production, the surface fine circuits can be laser etched with an optimized micro flow pressure, which solves the problems of local breaks or deformed circuits due to the conventional etching process and the structural layout.

## 1. Introduction

Tin-doped indium oxide or indium tin oxide (ITO) is the most important transparent conductive oxide (TCO) used as transparent electrodes due to its excellent electrical conductivity and optical transparency in the visible spectrum [1,2]. It is used in optoelectronic applications such as displays, touch panels, solar cells, and sensors [3,4]. In recent years, ITO glass has been increasingly used in the fields of banking equipment, medical equipment and all-in-one (AIO) computers [5,6]. The surface of the ITO glass is evenly distributed with circuits, which are prone to line sticking and short circuit due to the instability of the processing, thereby leading to a decrease the yield rate. In addition, ITO glass is a hard and brittle material, which makes it difficult to process.

In order to achieve precision etching, the ITO glass is generally manufactured using chemical etching, laser etching, and other processes. Bhosale et al. [7] presented an electrochromic film of WO_3_ fabricated on an ITO via a wet etching process using hydrochloric acid (HCl) for potential applications in energy-saving smart windows. Yang et al. [8] reported a fume etching process for ITO-glass with atmospheric fume of hydrochloric acid (HCl) that potentially opens a way toward rapid, nondestructive and quantitative bioanalysis. Mammana et al. [9] described a study on wet etching of thin films of indium tin oxide (ITO) using a simple method by monitoring the resistance of the thin film in aqueous solutions of oxalic acid and hydrochloric acid. Zhou et al. [10] achieved nano-modified ITO glass substrate with integrated internal and external outcoupling structures via wet chemical etching. Chen et al. [11] proposed nano-second pulse laser ablation-chemical etching (LACE) to prepare microstructures. Due to the limitation of the reaction rate, it requires s a long production cycle, and the environmental pollution of the chemical solution is inevitable in manufacturing ITO glass circuits.

Laser etching technology is expected to be applied in industrial products. With the increasing demand for promoting processing accuracy and minimizing processing damage, as well as the emergence of new processing requirements such as the yield of ITO glass, such processing methods have been developed [12,13]. The ultrafast femtosecond laser processing is a promising machining technique that minimized thermal damage [14,15] and generated a high material removal rate [16]. On the other hand, the photon energy of an ultraviolet (UV) laser is extremely high, which can destroy the chemical bond in the material molecule, thus directly ionizing it into plasma. Therefore, a UV laser usually has stronger ability to remove material, a more extreme processing scale, and less thermal damage when processing materials [17]. However, in the laser double-sided etching of ITO glass, the surface flatness is difficult to control due to workpiece positioning and laser thermal expansion.

For improving the flatness in the laser etching process, Yatsui Takashi et al. [18] developed a new surface flattening method that involved near-field etching, in which optical near-fields (ONFs) acted to dissociate the molecules. Yun et al. [19] proposed a virtual point removal algorithm for LS3DPCs with multiple glass planes. Weigel et al. [20] etched deep and narrow trenches on ultra-low expansion (ULE) glass with high anisotropy, which supported a prospective implementation. Rayerfrancis et al. [21] deposited an aluminium-doped zinc oxide (AZO) layer on etched glass and studied the surface roughness and optical properties. The surface flatness in the above methods is improved but the complexity of additional devices is difficult to control, and multi-step adjustment is required.

The microporous ceramics have been applied in the field of vacuum adsorption, precision filtration system, and non-contact product transmission. Wang et al. [22] provided the fundamental reference and theoretical basis for the design and optimization of textured bearings. Li et al. [23] studied the self-organized patterns of microporous ceramics, which were characterized by an arbitrarily functional circuit design and precise fixation. Yuan et al. [24] provided results of the preparation and filtration application of CNTs on porous materials. Zhou et al. [25] found that a microporous ceramic filter membrane had a large change in permeate flux during the initial stage of filtration and gradually tended to be stable. Although microporous filtration technology has been widespread, microporous ceramics have hardly been used in precision machining.

To determine the functional characteristics of the air flow of the thermal flow in microporous ceramics, Prasad et al. [26] studied the transient response characteristics and sensor signals of microporous layers and found reversible resistance changes of CO and H_2_. Simonenko et al. [27] investigated the effects of subsonic and supersonic dissociated air flow on the surface of microporous materials to determine the heating characteristics. Xiao et al. [28] studied a microporous ceramic membrane condenser (MCMC) with an average pore size of 3 µm for water and heat recovery from flue gas. Soloveva et al. [29] proposed a method of heat transfer intensification in open cell foam materials and studied the promising effect of micropores on the hydrodynamics and heat transfer. However, the dynamics of the airflow through microporous ceramics has not been studied preciously.

Based on traditional laser etching, the micro-airflow array pressurization is proposed by using micro-flow air bearing through airflow positioning. The objective was to achieve high-precision laser etching via pressurized micro-deformation of ITO glass during positioning. The micro-air flow and pressurized micro-deformation were first modeled in relation to the airflow pressure and etching gap in order to analyze the flatness variation behavior. Then, the surface flatness was investigated in relation to the airflow parameters and relative bearing location. Finally, it was applied to industrial ITO glass etching.

## 2. Laser Etching with Micro-Flow Air Bearing

### 2.1. Model of Laser Etching with Micro-Flow Air Bearing

Figure 1 shows the scheme of the ITO laser etching with a micro-flow air bearing. The traditional etching process involves laser micro-machining to directly etch ITO glass. However, the material’s thermal expansion leads to a decrease in workpiece flatness. In the developed process proposed in this study, a micro-hole ceramic body is used as a micro-air bearing, and the axis is aligned with the focused light beam. By loading pressurized gas (airflow pressure *p*) in the micro-hole ceramic body and adjusting the etching gap *h* between the ceramic body and ITO glass, the gas flows through the micro-hole array to the workpiece surface and imposes a certain air pressure on the surface. The micro-deformation generated by the micro-stress during the micro-airflow array pressurization is able to compensate the initial flatness of the workpiece within certain pressurization parameters. This can improve the flatness of the ITO glass during the etching process, making it more advantageous for surface positioning.

During the micro airflow array pressurization, the airflow generated by the airflow pressure *p* and the etching gap *h* after passing through the micro-pore will cause a maximum pressure *p_i_*, as shown in Figure 1. According to Bernoulli’s principle, the maximum pressure *p_i_* acting on the surface of the workpiece will result in micro-deformation of the workpiece, and the maximum depth of the micro-deformation is defined as Δ*s*. The workpiece also has a certain degree of error during positioning, which leads to an initial flatness *PV*_0_ due to its micro deformation. The micro-deformation Δ*s* of the workpiece contributes to the initial flatness *PV*_0_, and the compensation of this micro-deformation on the initial flatness *PV*_0_ is defined as the calculated flatness *PV_m_*. It can be expressed as follows:(1)$$\frac{1}{2}mv_{m}{}^{2}+p_{i}V=F\Delta s$$
(2)$$PV_{m}=\left|PV_{0}-k_{s}\Delta s\right|$$
where *m* is the mass of air flow, *v_m_* is the maximum airflow velocity, *V* is the volume of air flow, *F* is the force exerted on ITO glass, *PV*_0_ is the initial flatness, Δ*s* is the maximum micro-deformation, *k_s_* is the micro-deformation coefficient, and *PV_m_* is the calculated flatness.

The final surface flatness *PV* produced after laser etching is expressed as follows:(3)$$PV=k_{p}PV_{m}$$
where *PV* is the surface flatness, and *k_p_* is the flatness coefficient.

The combined equation of Equations (1) and (2) can give the formula for final surface flatness *PV*:(4)$$PV=k_{p}\left|PV_{0}-k_{s}\Delta s\right|$$

### 2.2. Structural Design of the Micro-Flow Air Bearing

Figure 2 shows the structural design of the micro-flow air bearing. According to the Bernoulli’s principle shown in Figure 2a, if the fluid velocity is high, the pressure is low in the air flow. The air flow velocity on both sides of the workpiece is different, resulting in the different pressure on both sides. In Figure 2b, the structure device consists of a micro-pore ceramic body, pneumatic cavity, gas pipe, frame components, balance screws, and quick connector components. The micro-pore ceramic body is the main core component of micro-flow air bearing. Its working principle is that pressurized air enters the pneumatic cavity through the gas pipe, the pneumatic cavity maintains a constant pressure, and the pressure outlet on the bottom surface of the pneumatic cavity supplies a constant pressure to the micro-pore ceramic body. The constant pressure produces a micro-pressure air flow acting on the surface of the ITO glass through the micro-pore in the ceramic. Thus, the micro-flow air bearing creates a certain pressure on the surface of the ITO glass, bringing the ITO glass close to the ideal flatness, which is beneficial for the double-side laser etching process.

## 3. Simulation, Experiment and Measurement Procedure

### 3.1. Simulation of Micro-Flow Air Bearing Pressurization

In laser etching, the flatness *PV* of ITO glass surface is dominated by micro-deformation by air flow pressure, which was modelled as shown in Figure 1. In order to analyze the air rheological behavior and workpiece micro-deformation using micro-flow air bearing, the finite element simulation was used in ANSYS v19.0 (Pittsburgh, USA). In the simulation of micro airflow array pressurization, the use of ANSYS facilitates the rapid and accurate calculation of the flow and pressure fields as well as micro-deformation on the machined surface. However, its computational efficiency is low when dealing with non-linear problems. The size of the micro-flow air bearing and ITO glass was consistent with the experiment. The pressurization process of workpiece was simulated with a three-dimensional model. For cold pressing simulation at room temperature of 25 °C, the workpiece was modeled as an elastic-plastic body and the influence of the micro-pore array on gas flow was simplified to describe using wind resistance coefficients in different directions. Other simulation conditions were consistent with experimental etching conditions as shown in Table 1.

### 3.2. Laser Etching Experiment of ITO Glass

Figure 3a illustrates the installation of micro-flow air bearings in laser etching equipment. The pressurized gas was supplied to the air bearings through a gas pipe, resulting in micro-air flow. The parallelism between the ceramic surface and ITO glass surface was verified using balancing screws. The etching gap was determined via the combination of a micrometer and thickness gauge. Double-sided laser etching could be performed after equipment process confirmation. In the double-sided etching of ITO, micro-flow air bearings were employed to optimize the support platform structure for product layout, material cost reduction, and design size. Figure 3b displays a comparison of the support platform before and after optimization. The design and sample size of the double-sided etching device with micro airflow array pressurization was limited. In the optimized platform, the supported beam was removed, for positioning of the micro airflow bearing and sample.

Thus, the optimization of the support platform effectively addressed the above-mentioned issues in double-sided etching.

Figure 3c displays the light path diagram of the double-sided laser etching of ITO glass. The etching laser was generated from the laser generator and traveled through the waveplate, beam expander, and other components. The focused etched spots were formed via the scanner on the surface of the ITO glass. The ITO glass workpiece was positioned on the lower equipment platform, using a processing size within the 400 × 500 mm range.

Table 2 shows the ITO glass etching parameters. The defocus of the airflow pressure *p* (0.16–0.19 kPa) and the etching gap *h* (1.7–1.9 mm) in the experimental parameters used after the application of micro-flow air bearings were −0.1 to 0.1 mm, indicating that the laser etching process window met the requirements.

### 3.3. Micron-Scale Flatness Measurement of ITO Glass

Flatness of ITO glass is a crucial parameter in the etching process. Figure 4 presents a schematic diagram of the method for measuring the surface of the ITO glass. In the measurement process, the measurement laser was emitted from the receiver, transmitted through the objective lens, and directed onto the surface of the ITO glass (Figure 4a). The laser spot diameter was 50 μm. The reflected laser from the surface finally reached the transmitter, resulting in a Gaussian distribution of the light intensity signal. The difference between the peak value and the substrate value represented the local deformation Δ*s* at the measurement point. The KEYENCE LK-H055 instrument was utilized to assess the flatness of the etched ITO glass (Figure 4b). The distance between the transverse and longitudinal measurement points was 45 mm and 30 mm, respectively, with 8 and 16 measurement points chosen on one side (Figure 4c).

Before the application of the micro-flow air bearings, the flatness contour detection of the ITO glass was depicted as shown in Figure 5. It can be observed that after the ITO glass was attached to the equipment platform, the surface *PV* deviated 80 μm horizontally and vertically.

In this study, a simplified model of air flow between the micro-flow air bearing and the workpiece was developed in ANSYS. The surface morphology of the ITO glass and micro-pore ceramic was observed via scanning electron microscopy (SEM, Zeiss Merlin, Jena, Germany). The surface flatness of the ITO glass was analyzed using a flatness meter (KEYENCE LK-HD500, Osaka, Japan).

## 4. Results and Discussion

### 4.1. Morphology of Micro-Pore Array in Micro-Flow Air Bearing

Figure 6 shows the SEM morphology of microporous ceramics. Ceramic blocks were characterized by micropores with micron-level pits on the surface, and the average diameter of micropores was about 20 μm. Due to their density, high hardness and wear resistance (physical properties), as well as good thermal resistance, excellent mechanical strength, and good high-temperature and corrosion resistance, ceramic materials with the above application characteristics are suitable for installation in laser etching environments.

### 4.2. Flow Characteristic of Micro Airflow Array Pressurization

Figure 7 shows the micro-flow rate distribution and micro-deformation of the micro-flow air bearings during etching of the ITO glass. It can be seen that the pressure *p_i_* generated by the micro-flow air bearings acted on the ITO glass surface. When the *p_i_* = 0.16 kPa and *h* = 1.7 mm, the maximum speed of airflow *v_m_* could be up to 3.04 m/s. In three-point and one-line synchronous motion, according to the finite element analysis, the ITO glass micro-deformation was related to the position of air bearings. Figure 7 (right) shows that under the mechanism of micro-airflow, the ITO glass generated micro-deformation, and the maximum value of Δ*S* = 0.84 μm. At the center, the surface was slightly deformed to the maximum. Here, when the ITO glass was double-sided laser etched, the ITO glass etching area produced a micro-airflow *p_i_* due to the air bearings that eliminated the deformation caused by the vacuum adsorption of the surface of the ITO glass. The use of micro-air-flow air bearings during the double-sided etching of the ITO glass optimized the etching process by reducing deformation and achieving accurate positioning, leading to a cleaner and more efficient etching process.

Figure 8 shows simulation results versus the airflow pressure *p* and etching gap *h*. Figure 8a shows the influence of the airflow pressure (*p*) and etching gap (*h*) on the maximum pressure *p_i_* on the workpiece surface. From this, it can be seen that the increased airflow pressure *p* increased linearly with the maximum pressure *p_i_*. On the contrary, the *p_i_* increased with a decreased *h*. When *h* dropped from 1.9 mm to 1.7 mm, *p_i_* increased from 12.6 Pa to 18.2 Pa, which was an increase of 44.4%. When pressure *p* increased from 0.17 kPa to 0.19 kPa, *p_i_* increased from 12.4 Pa to 18.2 Pa, which was an increase of 46.7%. When the micro-pressure of the air bearings is too low, it was insufficient to bring the ITO glass close to the ideal plane. Conversely, too large of a pressure would also have a negative impact on the surface flatness of ITO glass. Therefore, according to the simulation results, the etching parameters were reasonably selected for laser double-sided etching of the ITO glass.

Figure 8b shows the impact of the airflow pressure (*p*) and etching gap (*h*) on the maximum airflow velocity (*v_m_*). It can be seen that as the airflow pressure (*p*) increased, the maximum airflow velocity *v_m_* linearly increased, the velocity increased from 2.78 m/s to 3.35 m/s. With the increase in the etching gap (*h*), the change in the maximum airflow velocity was not evident and the difference is small. The velocity of airflow would affect the air pressure exerted on the surface of ITO glass, thereby affecting the final planarity of the etching.

Figure 8c illustrates the influence of the air flow pressure (*p*) and etching gap (*h*) on the micro-deformation Δ*s* of the workpiece. It can be seen that as the air-flow pressure increased, the micro-deformation (Δ*s*) increased linearly, and conversely, as the etching gap increased, the micro-deformation decreased. When h decreased from 1.9 mm to 1.6 mm, Δ*s* increased from 0.78 μm to 1.03 μm, which was a 24.3% increase. When *p* increased from 0.15 kPa to 0.20 kPa, Δ*s* increased from 0.77 μm to 1.02 μm, which was a 24.3% increase. When the air bearing produces insufficient micro-pressure, it was not enough to offset the deformation caused by vacuum suction, resulting in deviations from the ideal plane. Conversely, if it was too high, it could also cause excessive deformation on the surface of the ITO glass.

### 4.3. Analysis of Circuit and Surface Micro-Deformation of ITO Glass

Figure 9 shows a comparison of the SEM morphology of laser double-sided etching lines of ITO glass with and without a micro air-flow air bearing after optimization. The etching line without the micro air flow air bearing was relatively poor compared to the etching line with the application of a micro air flow air bearing, including defective and deformed circuits (Figure 9a). According to the morphology, the occurrence of open or short circuits was mainly due to the lack of an effective guarantee of the flatness of the ITO glass surface. Furthermore, when comparing the maximum width of the etched lines, the maximum width was reduced from 5.76 μm to 5.30 μm, which represented a reduction of 8.07%. This indicated that the micro airflow array pressurization resulted in a flatter surface and reduced line distortion. Therefore, after applying the micro air flow air bearing, the flatness of ITO glass could be improved and the effect of the double-sided etching lines was better (Figure 9b).

According to the experiment, it was found that in production, commonly used ITO glass etching yielded a better surface flatness when the air flow pressure was around 0.18 kPa and the etching gap was around 1.9 mm. These engineering parameters were taken as the basis for studying the effect of the air-flow pressure and etching gap on glass flatness. Figure 10 shows the distribution of the surface flatness in laser etching with and without the application of a micro-flow air bearing. It can be seen that before the application of the micro-flow air bearing, the maximum and minimum values of the surface flatness differed greatly, while after the application of the micro air flow air bearing, the flatness distribution was more uniform.

According to Figure 10, when the airflow pressure *p* was 0.18 kPa, the maximum and minimum local flatness of the ITO glass workpiece without a micro-air bearing were 40 μm and −45 μm, respectively, with a total flatness of 75 μm. However, with the application of the micro-flow air bearing, the maximum flatness was 18 μm, the minimum was −4 μm, and the total flatness was 22 μm. Thus, the use of micro air bearings could significantly improve the flatness of the processed ITO glass surface.

### 4.4. Analysis of the Surface Flatness in Relation to Pressurization Parameter

Figure 11a shows the effect of different airflow pressure *p* on the flatness *PV* of the double-sided etched ITO glass with a micro-flow air bearing. It can be seen that when the airflow pressure *p* was in the range of 0.05 to 0.20 kPa, the *PV* of the ITO glass after applying the micro-flow air bearing was lower than the initial surface flatness. The flatness decreased first and then increased with the increase in airflow pressure. The available airflow pressure was between 0.11 and 0.19 kPa, where the flatness could be maintained within 30 μm, and it was the flattest at 0.16 kPa with a value of 24 μm, which was 70% lower than the initial workpiece surface. Therefore, it can be concluded that the use of a micro-flow air bearing to optimize the surface flatness of ITO glass was ideal with the optimized airflow pressure. The analysis also showed that the ideal airflow pressure *p* for the micro-air bearing was between 0.11 kPa and 0.18 kPa. The simulation results corresponded to a pressure range of 9.5 to 15.7 Pa on the workpiece surface, resulting in a high surface flatness after etching.

Figure 11b shows the effect of different etching gaps *h* on the flatness *PV* of the ITO glass after etching with the micro-flow air bearing. The etching gap increased from 0.5 mm to 2.0 mm by 0.05 mm each test. It can be seen that as the etching gap increased, the flatness also decreased first and then increased. When the etching gap *h* was 1.6 to 1.8 mm, the flatness was less than 30 μm. The optimal etching gap *h* was 1.6 to 1.8 mm, for which the flatness reached the lowest value of 22 μm. The results indicated that optimizing the etching gap in the micro-flow air pressurization could achieve a more ideal flatness *PV* of the ITO glass. The simulation results depicted in Figure 8 reveal that the maximum pressure *p_i_* exerted on the workpiece surface fell within the range of 12.9 to 15.2 Pa, with optimal flatness achieved after the etching process.

The above experiments showed that when the air pressure applied to the workpiece surface was within the range of 15.2 to 15.7 Pa, the surface planarity achieved after laser etching was at its lowest value. In inappropriate airflow pressure ranges, laser etching and the deformation of ITO glass could not balanced. When the air pressure wwas below the optimal range, the surface deformation was significant due to insufficient pressure to compress the surface (Figure 8a) when the pressure was above the range, excessive downward pressure is generated at the center, which caused bending (Figure 8a). Neither of these were not favorable for the application of the micro-airflow air bearing in the laser etching process.

### 4.5. Critical Values of Pressurization Parameter to Surface Flatness

Figure 12 shows the results for the critical airflow pressure *p_c_* versus the surface formation, which were theoretically calculated from Equation 3 via a data-twin method using the experimental data in Figure 11. It can be seen that the maximum surface flatness *PV* of the laser etching first increased then decreased with the increasing airflow pressure *p* and etching gap *h*. As the pressurization parameters continue to increase, *PV* first decreases to the initial flatness *PV*_0_, then decreased to zero (ideal flatness), and finally continued to increase.

When *PV* was less than 30 μm, it could be considered that high-quality ITO glass surface etching could be achieved. In this case, the critical airflow pressure corresponding to different gaps was defined as *p_c_*. The same etching gap *h* corresponded to two values of *p_c_*, which were both feasible airflow pressures *p* within this range. In theory, under an ideal airflow pressure *p_b_*, etching with a flatness of 0 μm can be achieved, but there still exists a certain amount of clamping error and air flow divergence, that cause errors between experimental values and theoretical calculation values. In summary, this theoretical model can provide an optimized process window for ITO double-sided etching, thereby improving the etching quality and efficiency.

### 4.6. Application of Micro Airflow Air Bearing in Laser Etching

In industrial laser etching of ITO glass, a supported beam is installed on the platform in order to maintain the workpiece flatness. However, improvement is limited and the flatness is different to control. In the developed process proposed in this study, a micro-airflow array pressurization was applied during laser etching. The advantage was that the micro-deformation generated by the micro-stress during micro airflow array pressurization was able to control the pressurization parameter.

Based on the above experiments, the optimal parameters were selected for laser etching of the ITO glass circuit. Figure 13 shows the morphology of the etched ITO surface and a silver paste circuit using a 20× lens of an optical microscope, with circuit widths of 8 μm and 25 μm, respectively. The results showed that the circuit lines were uniform and clear with excellent quality, indicating that after applying the micro-flow air bearing, the etched circuits were uniform and rarely showed broken or open lines, which was beneficial for production and improved production product quality.

## 5. Conclusions

Based on traditional laser etching, micro-airflow array pressurization was proposed by using a micro-flow air bearing through airflow positioning to achieve high-precision laser etching by pressurized micro-deformation of ITO glass during positioning. In industrial production, the surface fine circuits can be laser etched with optimized micro-flow pressure, which solves the problems of local breaks or deformed circuits due to the conventional etching process and the structural layout. Some key conclusions from this study were as follows:A micro airflow array pressurization was proposed in a double-sided laser etching process by using a micro-airflow air bearing. The surface underwent positive pressure by loading the airflow pressure and adjusting the etching gap to position the machined surface. The micro-deformation generated by the micro-stress was able to compensate the initial flatness of the workpiece within critical pressurization parameter. This was beneficial for improving the flatness of the ITO glass during the etching process.The application of micro-airflow array pressurization enhanced the uniform pressure distribution on a large area of the glass surface, thus facilitating the positioning of the etched surface. The maximum positive pressure was achieved under the micro-airflow air bearing and the maximum positive pressure linearly depended on the air flow pressure and the etching gap. Insufficient pressure was not enough to press down the surface, while an excessive pressure would cause excessive micro-deformation in the processing area, leading to incompleteness of the final etched lines.The airflow pressure and etching gap showed a trend of first decreasing and then increasing with respect to the workpiece flatness. Within the critical airflow pressure range, the pressure applied on the surface of the workpiece was sufficient to hold it down without causing excessive micro-deformation. When the airflow pressure was in the range of 0.11–0.18 kPa and the etching gap was in the range of 1.6–1.8 mm, the optimal pressure on the workpiece surface was 15.2–15.7 Pa and the flatness after etching is 22 μm, which was a decrease of 74%.The application of a micro-airflow air bearing could address the key issues of planarity and processing efficiency of the etched surface. The resulting circuit line distribution was more uniform. It solved the problems of local micro-circuit short-circuits or open-circuits caused by traditional etching, thereby improving the product yield. In addition, the static electricity problem in product etching was avoided.

## Figures and Tables

**Figure 1 micromachines-14-00676-f001:**
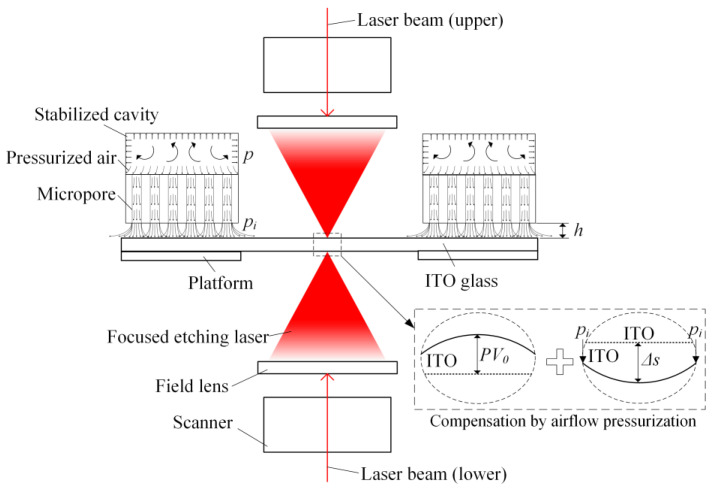
The scheme of ITO laser etching with micro airflow array pressurization.

**Figure 2 micromachines-14-00676-f002:**
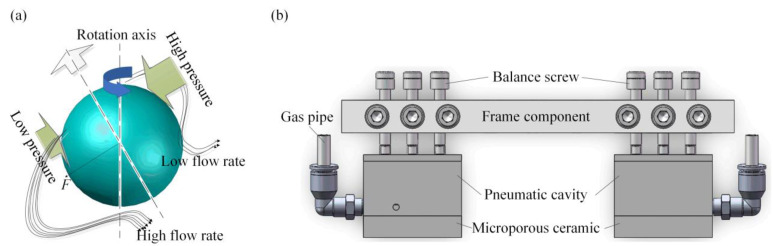
Structural design of the micro-flow air bearing: (**a**) bernoulli principle in airflow array pressurization; (**b**) structure of micro-flow air bearing device.

**Figure 3 micromachines-14-00676-f003:**
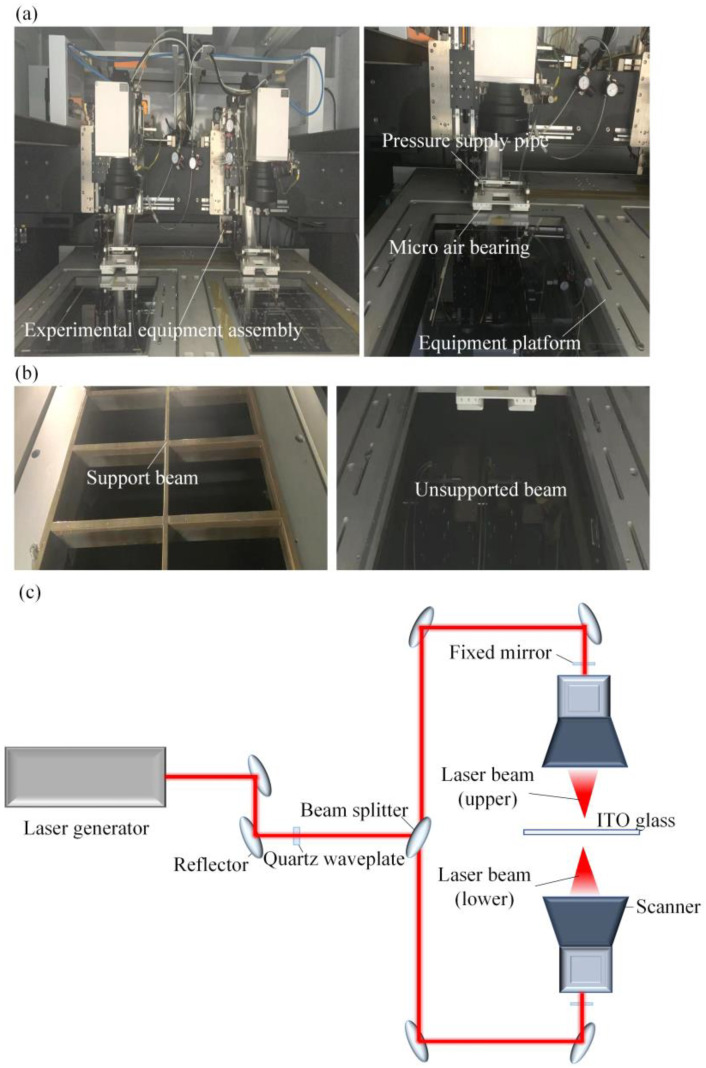
Experimental device of ITO laser etching: (**a**) experimental device assembly of laser etching; (**b**) structure optimization of support platform; (**c**) laser light path.

**Figure 4 micromachines-14-00676-f004:**
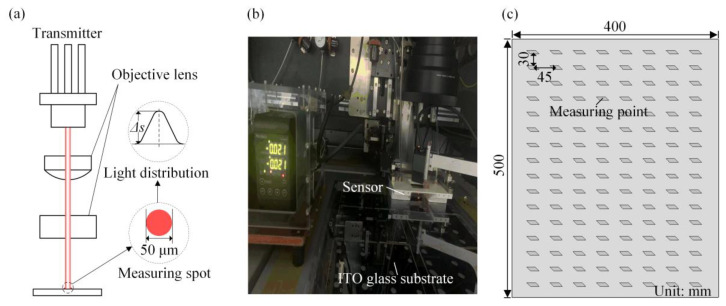
Surface flatness measurement of the ITO glass: (**a**) scheme of measurement principle; (**b**) measurement device and sensor; (**c**) measuring point distribution.

**Figure 5 micromachines-14-00676-f005:**
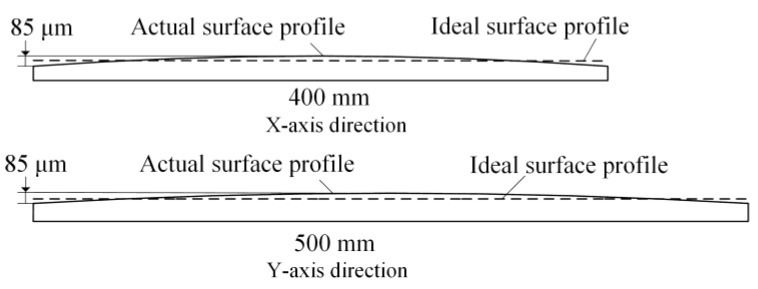
Scheme of micro-deformation of ITO glass.

**Figure 6 micromachines-14-00676-f006:**
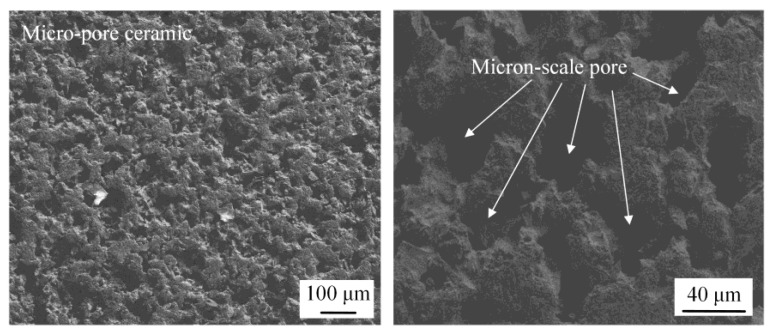
SEM morphology of micro-pore array on microporous ceramic surface.

**Figure 7 micromachines-14-00676-f007:**
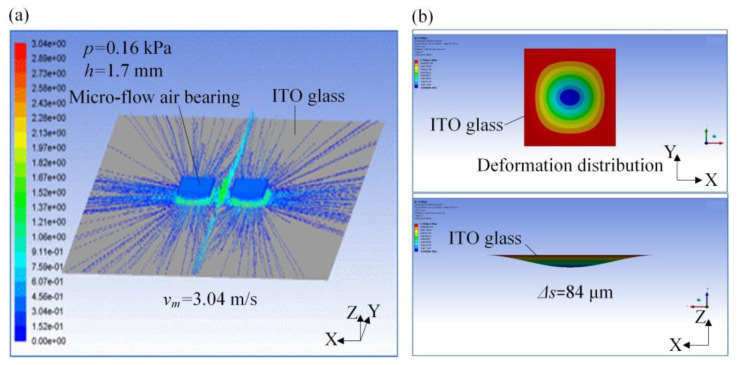
Micro-flow rate distribution and micro-deformation of micro airflow array pressurization: (**a**) micro-flow rate distribution; (**b**) micro-deformation.

**Figure 8 micromachines-14-00676-f008:**
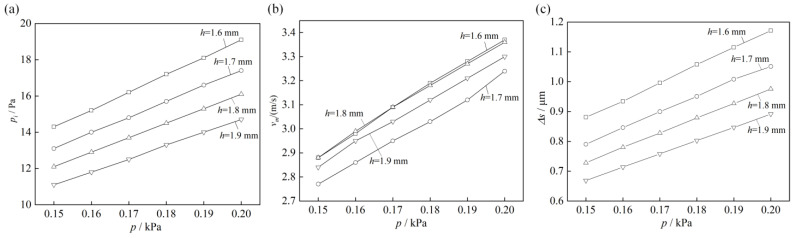
Simulation results versus airflow pressure *p* and etching gap *h:* (**a**) maximum pressure *p_i_*; (**b**) maximum airflow velocity *v_m_*; (**c**) maximum micro-deformation Δ*s*.

**Figure 9 micromachines-14-00676-f009:**
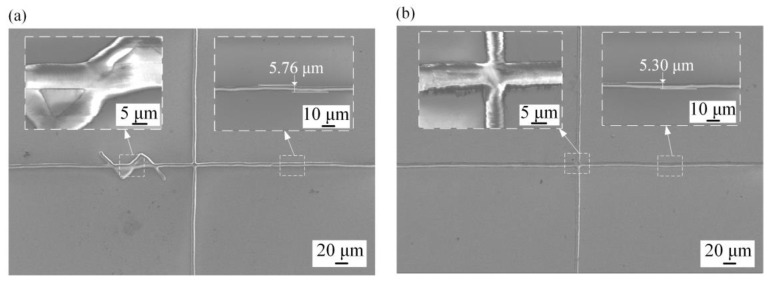
SEM morphology of circuit line on ITO glass: (**a**) without pressurization; (**b**) with pressurization.

**Figure 10 micromachines-14-00676-f010:**
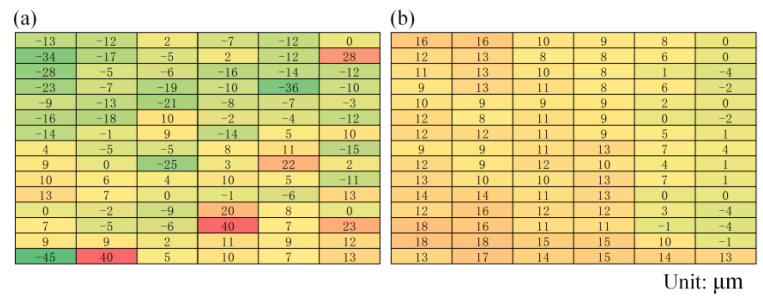
The flatness distribution of ITO glass: (**a**) before pressurization; (**b**) after pressurization.

**Figure 11 micromachines-14-00676-f011:**
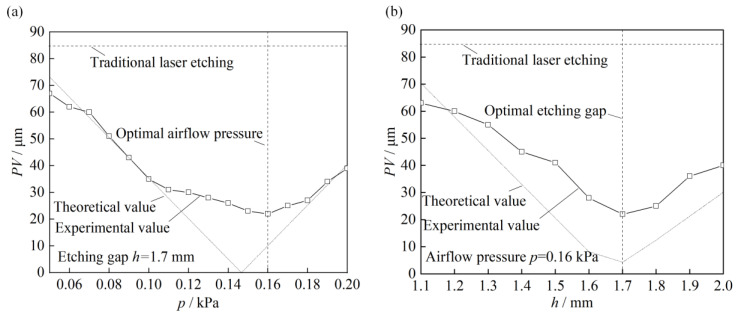
Maximum surface flatness *PV* versus (**a**) airflow pressure *p* and (**b**) etching gap *h*.

**Figure 12 micromachines-14-00676-f012:**
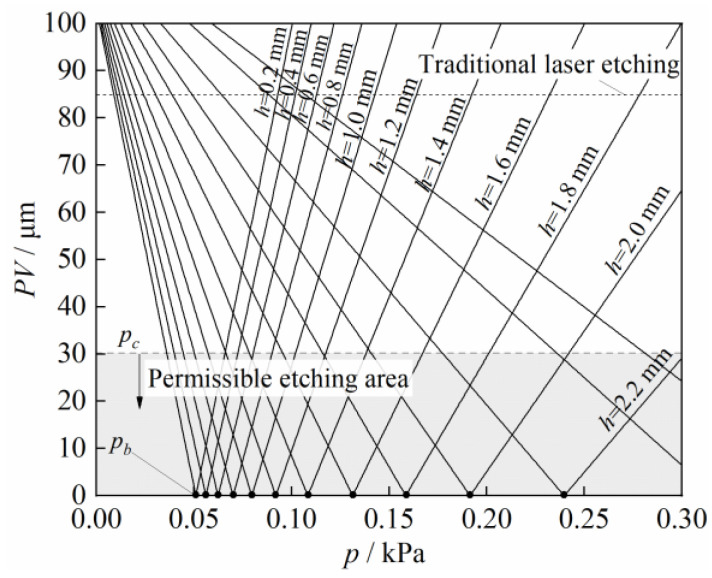
Maximum surface flatness *PV* versus critical values of critical airflow pressure *p_c_*.

**Figure 13 micromachines-14-00676-f013:**
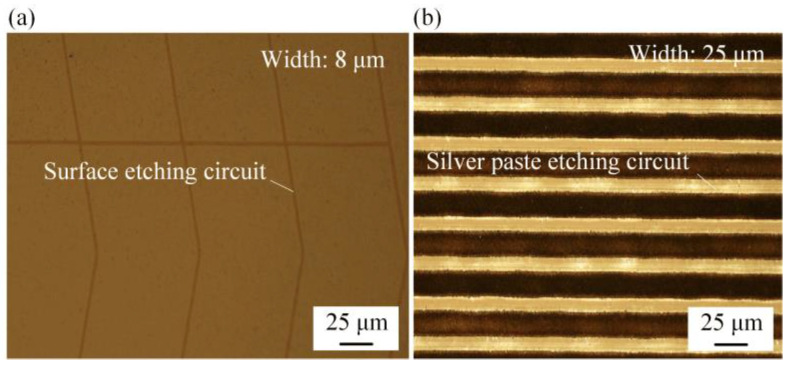
Morphology of surface etched circuits: (**a**) ITO glass surface circuit; (**b**) silver paste circuit.

**Table 1 micromachines-14-00676-t001:** Simulation conditions of micro airflow array pressurization.

Simulation Parameter	Value
Airflow pressure *p* (kPa)	0.16–0.19
Etching gap *h* (mm)	1.70–1.90
Micro-pore viscosity resistance (1/m^2^)	Horizontal direction: 2.111 × 10^5^ Vertical direction: 2.111 × 10^8^
Micro-pore inertial resistance (1/m^2^)	Horizontal direction: 1 × 10^3^ Vertical direction: 2.4 × 10^5^

**Table 2 micromachines-14-00676-t002:** Experimental parameters of ITO glass etching.

Experimental Parameters	Parameter Value
Laser power (W)	ITO glass: 0.40 ± 0.05 Sliver paste: 0.70 ± 0.05
Laser frequency (kHz)	280
Laser wavelength (nm)	355
Defocus amount (mm)	−0.1–0.1
Feed speed (mm/s)	ITO glass: 700 ± 100 Sliver paste: 600 ± 100
Scanner calibration (μm)	≤5
Workpiece thickness (mm)	0.55
Maximum etching depth (Å)	135
Etching mode	Double side

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
