# Peer review of "Improving Etched Flatness by Micro Airflow Array Pressurization in ITO Glass Laser Machining"

_micromachines, 2023, doi:10.3390/mi14030676_

Round 1

Reviewer 1 Report

Remarks

1. Laser etching of materials has been known for quite a long time. What is the advantage of the proposed laser etching technology compared with existing ones?  What is the scientific novelty of the research? Please specify in the text of the article.

2. What are the advantages of the developed laser etching technology over the ones already existing and in use in industry? Please specify in the text of the article.

3. Does the laser etching of materials proposed by the authors provide a stress concentrator? Have residual and other stresses in the material after laser etching been investigated?  Specify in the text of the article.

4. What is the maximum depth of material etching, specify in the text of the article.

5. What is the technological novelty of the research, specify in the text of the article.

6. In what areas of industry can the materials processed by the proposed technology be used? Specify in the text of the article.

7. What is the optimum pore size after laser etching? What is the effect of porosity on the properties of materials after laser treatment. Indicate in the text of the article.

Reviewer 2 Report

Authors developed a setup employing micro-airflow for substrate bending compensation during laser etching process. They applied it on ITO glass and demonstrated an improvement of post-processing flatness comparing to standard laser etching without subsrate bending compensation. The article needs a minor revision, some information is missing and language needs improvement. Below are detailed comments.

1) Line 34: please specify what "AIO" acronym stands for.

2) Line 35: could you please explain "which are prone to adhesion due to the instability of the processing"? Is there a word missing? Adhesion failure?

3) References 7-11 are irrelevant in the introduction part. Authors are supposed to show ITO etch progress done with different patterning methods, but instead they refer to a mix of etch technologies on very different materials, except ITO.

4) Line 65: please specify what "ULE" acronym stands for.

5) Line 140: supposed to be ANSYS I guess?

6) Line 141: sentence "The etching process of workpiece 141 was simulated with three-dimensional shape" needs to be re-written for better clarity.

7) Line 147: Table 1 lacks units.

8) Line 156: authors mention support platform was optimized for double-side etching, but it is not clear from text and Figure 3b what exactly was optimized, and why.

9) Figure 3c: seems that some elements have incorrect labels, please check beam splitter and beam expander.

10) Line 173: "requirement" should be replaced with "parameter".

11) Line 255: authors compare etched line quality with and without micro-airflow, claim line quality is poor without micro-airflow. This is not enough to evaluate improvement, some quantitative metrics should be applied, like linewidth variation and roughness. Also, in Figure 9a, they focus on a feature that looks like manufacturing defect, not sure the probability of such defect occurence is lower with micro-airflow.Maybe authors have collected more statistics that they could show?

12) Authors need to benchmark their method against other solutions existing in industry.

Reviewer 3 Report

In this paper, by means of simulation and experimental verification, the author uses the micro-flow air bearing located by air flow to carry out array pressurization, studies the relationship between surface flatness, air flow parameters and relative bearings, and realizes high-precision etching through the micro-deformation of ITO glass under pressure. The logic is complete, the language is simplified, the graphic is beautiful, and the workload is large. However, the following problems need to be pointed out:

1. The literature review in the second paragraph of the introduction is not clear. It does not clearly state what specific problems they have studied and what contributions they have made.

2. In the introduction, the author has carried out a large number of reviews on laser etching technology, but the aspect of microfluidic air is rarely mentioned. It is suggested to increase the number and content depth of literature.

3. In 3.1, there is a lack of advantages and disadvantages of ANASYS compared with other software.

4. In Figure 3 (a) and (b), the overall light is dim, and in a, the marked part in the left figure is close to the blue pipe, which is easy to cause visual deviation. It is suggested to mark the position reasonably.

5. In Figure 5, the mark line of ideal surface contour in the following figure is too short and not significant enough, so it is suggested to reasonably extend it.

6. As shown in Figure 9, the background color is dark and the box lines lack brightness in the figure, it is suggested that the box lines and lines should be marked with reasonable color matching.

7.  It is mentioned in the 4.3 end paragraph that the maximum local flatness of ITO

 glass workpiece without micro-gas bearing is 30μm, but the data in Figure 10 shows that the maximum flatness should be 40μm. Is there any data error?

8. Direct expression of purpose and emphasis on the contribution of work are recommended in the conclusion.

Round 2

Reviewer 1 Report

All responses to comments should be transferred to the text of the article.  Response 1 and etc.